# Alcohol Consumption Frequency of Parents and Stress Status of Their Children: Korea National Health and Nutrition Examination Survey (2007–2016)

**DOI:** 10.3390/ijerph17010257

**Published:** 2019-12-30

**Authors:** Serin Kim, Wonjeong Chae, Seung Heon Min, Yerim Kim, Sung-In Jang

**Affiliations:** 1College of Medicine, Yonsei University, Seoul 03722, Korea; lynn307@naver.com (S.K.); alstmdgjs@naver.com (S.H.M.); chelseak@naver.com (Y.K.); 2Department of Public Health, College of Medicine, Yonsei University, Seoul 03722, Korea; wjchae0816@yuhs.ac; 3Institute of Health Services Research, Yonsei University, Seoul 03722, Korea; 4Department of Preventive Medicine, Yonsei University College of Medicine, Seoul 03722, Korea

**Keywords:** adolescent, stress level, parental drinking, drinking frequency, parenting, South Korea

## Abstract

*Background*: The effect of stress on mental health has been increasingly acknowledged. Drinking habits are closely inter-related with stress and each affects the other. However, only limited studies addressed the effects of alcohol consumption on family members apart from spouses. The purpose of this study is to better understand the relationship between parent drinking frequency and their children’s self-reported stress. *Methods*: Data was collected from the Korean National Health and Nutrition Examination Survey (K-NHANES) conducted during 2007–2016. Respondents were divided into three groups: children (*n* = 3796), maternal (*n* = 22,418), and paternal (*n* = 16,437). After merging the children and parents data sets, we identified the final study population of 3017 and performed binary logistic regression. *Results*: We found that the odds of high stress cognition was 1.58-fold higher for children who have heavy drinking mother (95% CI: 1.14–2.19) and 1.45-fold higher for those who have heavy drinking father (95% CI: 1.06–1.99). In a subgroup analysis, children whose household income level was Q1 and maternal occupation was white collar showed a statistically significant association of high stress with parental drinking frequency. *Conclusions*: Parental drinking frequency negatively impacts stress in the children of drinkers. We suggest providing support care for children in vulnerable environments to improve their stress levels.

## 1. Introduction

Excessive alcohol consumption has negative consequences drinkers as well as others and can lead to severe physical and mental disorders. Chronic disorders and diseases that are specifically related to the liver or stomach can occur from alcohol [1,2,3,4]. As per mental disorder, previous studies have investigated the linkage between alcohol consumption and mental disorder [5,6]. Also, excess alcohol consumption affects peers negatively that heavy drinkers might face issues with family members and friends [7,8]. When the heavy alcohol consumption occurs, it affects the children of heavy drinkers the most since they are not able to receive necessary care from parents [9,10,11].

High alcohol consumption is a public health problem that the World Health Organization (WHO) has attempted to address [12]. According to the 2008–2010 Korea Center for Disease Control and Prevention, the monthly alcohol consumption rate has increased from 54.1% to 58.2% in 4 years [13]. Moreover, the rate of people with drinking problems has also increased [13]. In Korea, men have higher consumption of alcohol than women due to cultural factors [14,15,16] and recently, women’s alcohol consumption has also increased [17]. The most recent data from Korean Statistics show that the consumption has increased from 73.5% (2007) to 75.0% (2017) for men and 41.5% (2007) to 50.5% (2017) for women [18].

Behavior associated with alcohol consumption is closely associated with stress [19,20]. Alcohol consumption and stress levels affect each other. Many studies show that drinking is a stress relief factor [19] and occupational stress can lead to excessive drinking for Korean workers. On the other hand, drinking can have a detrimental effect on the ability to cope with stress [20]. Korea ranked 17th place in total alcohol consumption over the age of 15 in liters per capita on the WHO Global Status report in 2014. Korea consumes 12.3 L per year, which was the highest in Asia [21]. The WHO health statistics for 2016 reported that the global average of total alcohol per capita over the age of 15 was 6.4 L per year and the regional average was 7.3 L per year. Again, Korea was the highest alcohol consuming country in Asia, at 10.2 L per year [22]. As the data has shown, Korea has a major problem related to heavy alcohol consumption that needs to be studied in various aspects, including mental health and its consequences.

Culturally, Koreans tend to drink in groups, rather than solitarily, and this is closely associated with social and work environment. There is the so-called “work-dinner” which can be described as a team dinner after work which involves drinking [17,23,24]. During this dinner, workers are obligated to drink with their colleagues and this leads to a high volume of alcohol consumption. This unfortunate custom occurs regardless of occupational groups and gender [17,25]. Previously, we were focusing on alcohol-related problems among labor workers. A recent study discovered that 64% of high alcohol-consuming disciplines are office workers such as directors and managers [26].

Many studies have aimed to understand the effect of stress on physical and mental health [27,28]. For example, stress can contribute to diseases such as cancer, and mental health problems, such as anxiety disorders, burnout, depression, and PTSD among adults [29,30]. However, stress is not a problem only apply to adults; adolescents also suffer from the negative impacts of stress such as anxiety and poor cognitive development [31,32,33]. Many factors can cause stress in adolescents, such as friends, school, and family. In Korea, one of the biggest reasons for stress is related to academic achievement, largely due to the parental pressure on children, which is even more influential than the child’s frustration with his/her academic results [34,35,36]. As parents have a major influence on their children’s stress level and a good relationship between parents and children would be beneficial. Therefore, society should encourage families to build good relationships. One strategy to build good family relationships would be to decrease parental alcohol consumption. Children with parents who consume a high volume of alcohol suffer from high-risk environments and poor health status including development delay, social isolation, and increased risk-taking behaviors [37,38,39,40]. However, among various mental health problems, there was no recent study on the direct association between children’s stress and parental alcohol consumption since 1988 [41]. As stress can cause mental health problems [31,37,42], it is important to control the risk factor for children especially those who are living with heavy alcohol consuming parents. This emphasizes the need to identify and resolve the stress factors for children.

Therefore, in this study, we investigated the association between the frequency of parental alcohol consumption and children’s stress level using the Korean National Health and Nutrition Examination Survey (K-NHANES).

## 2. Methods

### 2.1. Study Design and Participants

The data used for this study were taken from the K-NHANES from 2007 to 2016. We divided the raw data into three data sets: children, adult women, and adult men. Subjects between 12–19 years old were defined as children Subjects over the age of 30 were divided into adult women and adult men according to their gender. Then, maternal and paternal variables were created by matching adult women and adult men data to the father and mother ID of children. The data for the children of a single parent was removed. After the selection, we defined adult women who matched 3017 children as ‘maternal’ and adult men as ‘paternal’.

### 2.2. Dependent Variable

The study’s dependent variable was children’s self-perception of stress. We asked subjects to grade their responses depending on how much stress they felt. To create a stress indicator, we divided the grading into two groups based on the survey questions and its response in K-NHANES: ‘very often’ and ‘often’ were defined as ‘yes’, and ‘a few times’ and ‘occasionally’ were defined as ‘no’. Our grouping method was followed by K-NHANES guidelines for stress status measurement.

### 2.3. Intersting Variable

The main independent variable was the parental alcohol consumption frequency. We analyzed maternal and paternal data separately. The survey asked subjects, “How often did you drink alcohol in the last year?” K-NHANES responded to the frequency of drinking into six scales: (1) never, (2) less than once a month, (3) once a month, (4) 2–4 times a month, (5) 2–3 times a week, and (6) more than 4 times a week. To detect significant differences in analysis, their responses were regrouped into three: None (never), Average (less than 4 times in a month and once a month), and High for the remaining categories.

### 2.4. Covariates

For children, we considered two main categories of confounding variables: demographics and health-related factors. Demographics included gender and age. We separated age according to education level (elementary school: 12, middle-school: 13–15, high-school: 16–18). Income level was grouped in quartile based on the response of the survey question. Health-related factors included depression, self-assessment of health, and physical activity (No: no physical activities; Minimum: walking or weight exercise 1–2 times a week; Average: walking or weight exercise 3–4 times a week; Maximum: walking or weight exercise 5–7 times a week). For the self-assessment of health, the original criteria created five response groups (very good, good, normal, bad, and very bad), which we converted into three groups (good, normal, and bad).

For parents, we considered three types of confounding variables: demographics, health-related factors, and socio-economic factors. For demographics, we used three age groups (30s, 40s, and over 50s). Health-related factors included smoking status (current smoker, non-smoker, and ex-smoker) and self-assessment of health. Socio-economic factors included academic background (elementary school, middle school, high school, and university), household income (four grades from high to low), and occupation (White collar: administrative and management personnel; Pink Collar: service industry workers; Blue Collar: manual labor industry workers; Others: student, homemaker, and unemployed).

### 2.5. Statistical Analyses

We first examined general characteristics in child and parent populations. Using a chi-square test and binary logistic regression, we also analyzed statistical significance with regard to the stress status for each variable and determined adjusted odds ratios and 95% confidence intervals for children’s stress status. We conducted the Hosmer and Lemeshow Goodness Test to evaluate the goodness of the binary logistic model. In the PROC LOGISTIC procedure, the Hosmer and Lemeshow chi-square tests implemented with the LACKFIT option were used to test the suitability. The analysis was adjusted for the possible confounders. *p*-values < 0.05 were considered statistically significant. All statistical analyses were performed using SAS version 9.4 (SAS Institute, Inc., Cary, NC, USA).

### 2.6. Ethical Statement

The study used data from the Korea Centers for Disease Control and Prevention as open data that any researcher can utilize for their study to enhance the health of the Korean population. The survey was approved by the Institutional Review Board (IRB) of the Korea Centers for Disease Control and Prevention.

## 3. Results

Table 1 shows the characteristic of the study subjects (*N* = 3017). Of the total, 52.7% (1589) children are boys and 47.3% (1428) are girls. We found that 754 children (25.0%) reported that they felt stressed. The difference in the level of stress felt between a boy and a girl was not statistically significant. Of mothers, 25.9% (780) are under the ‘none’ category, 62.9% (1899) and 11.2% (388) are under the ‘average’ and ‘high’ categories respectively. Of fathers, 12.2% (367) are under the ‘none’ category, 43.7% (1319) and 44.1% (1331) are under the ‘average’ and ‘high’ categories respectively. Overall, fathers drink more intensively; 44.1% are in the ‘high’ category compared to 11.2% of mothers.

Table 2 shows the results from binary logistic regression analyses of the associations between parental drinking frequency and stress in children. Intensive paternal drinking was associated with high stress cognition in children. We observed a 1.58-fold higher risk of high stress cognition in children associated with high maternal drinking frequency (95% CI: 1.14–2.19) and a 1.45-fold higher risk associated with high paternal drinking frequency (95% CI: 1.06–1.99) compared to children whose parents do not drink. Children who experienced average parental drinking frequency had increased risk of stress (maternal: OR = 1.39, 95% CI: 1.11–1.74; paternal: OR = 1.21, 95% CI: 0.89–1.66) compared to children of parents who do not drink. We found that health-related and socioeconomic factors of the children, including depression (yes, OR = 2.36, 95% CI: 1.81–3.08), and self-assessment of health (normal: OR = 1.46, 95% CI: 1.21–1.77) and bad: OR = 2.00, 95% CI: 1.40–2.87), income (Q2: OR = 0.74, 95% CI: 0.56–0.98) were significantly associated with high stress cognition in children.

Table 3 and Table 4 shows the results from subgroup analysis of the associations between high stress cognition and parental alcohol consumption frequency based on children’s gender, age group, household income and parent’s occupation.

For maternal drinking frequency, we observed statistically significant odds ratios for the following variables in children. Gender (Boys: average: OR = 1.39, 95% CI: 1.02–1.88; high: OR = 1.45, 95% CI: 0.99–2.29); Girls: average: OR = 1.44, 95% CI: 1.03–2.00; high: OR = 1.77, 95% CI: 1.09–2.89), household income (Q1: average: OR = 2.80, 95% CI: 1.61–4.87; high: OR= 4.05, 95% CI: 1.69–9.74), and children whose maternal occupation (White collar: average: OR = 1.69, 95% CI: 1.17–2.44; high: OR = 2.38, 95% CI: 1.42–3.98) (Table 3).

For paternal drinking frequency, we observed statistically significant odds ratios for the following variables in children. Gender (Boys: average: OR = 1.42, 95% CI: 1.12–1.80; high: OR = 1.16, 95% CI: 0.42–3.25; Girls: average: OR = 1.42, 95% CI: 1.12–1.80; high: OR = 1.77, 95% CI: 1.15–2.31), household income (Q1: average: OR = 2.46, 95% CI: 1.15–4.00; high: OR = 2.77, 95% CI: 1.30–5.91), and children whose parental occupation (White collar: average: OR = 1.30, 95% CI: 0.94–1.80; high: OR = 2.16, 95% CI: 1.34–3.47) (Table 4).

## 4. Discussion

Our study investigated the relationship between children’s stress status and parental drinking frequency. To find a link between two factors, we used binary logistic regression models. An analysis of the maternal and paternal drinking frequency showed that the child was more affected by the mother than father. While fathers showed a significant correlation only in groups with ‘high’ drinking frequency, mothers in both ‘average’ and ‘high’ groups negatively affected their children’s stress.

After the main analysis, we performed subgroup analyses for specific variables (sex, age, household income, and maternal occupation) to identify significant correlations for specific groups. The result of the main analysis showed that both parent’s alcohol consumption can cause stress on their children. Between mothers and fathers, mothers have a greater effect on children which is similar to previous studies on the effect of mother–child relationships on children [43,44,45]. The studies reported that mother–child relationship quality can impact the child’s self-esteem, social competence, and mental health [43,44,45]. We investigated further through subgroup analysis to discover other risk factors that can influence the outcome. As our study was designed from the top-down model building strategy, we chose to see associations among socioeconomic factors and children’s factors. Significant results were obtained from the following three variables: sex, household income, and occupation of parents.

From the subgroup analysis, we found statistically significant gender differences at the child stress level depending on the frequency of parental drinking. Boys and girls had stress when their parents consume alcohol, yet girls showed higher OR and it was statistically significant. The results were in line with an earlier study that women were more vulnerable to stress than men [42,46] and our study supported the finding that it can be extended to adolescents. Between mothers and fathers, children were more affected by the mother’s drinking and this outcome can be also explained by the mother–child relationship [44,45,47].

We found that high school children were most affected by parental drinking frequency. We interpret this finding to mean that children of this age have their stress from factors such as university admission, friendship, and family matters. A previous study showed that high school years are a stressful stage for various reasons [32,48,49]. For Korean students, the biggest impact on stress is from academic pressure, which can lead to suicide [35,48,49]. Many studies have discussed how a lack of support from parents could increase suicide risk [48,49]. Therefore, we should have support programs for children who are living with parents with heavy drinking habits while seeking means to manage the stress level of children.

Also, in children from low-income families, the vulnerability of stress was evident. This was consistent with prior papers that say low-income people are more vulnerable to stress [42,47,50]. We found that in families that are not financially comfortable, children whose mothers drank more frequently were under more stress. Based on this, special support should be given for low-income households, especially families with parents who have high drinking and frequent drinking habits. Basic financial support, along with the chance of an abstention program with children’s mental health checkups to ease the child’s stress at home. However, we notice that even with the household income being high, children are under stress when their parents consume a high level of alcohol. This explains that parent’s alcohol consumption has a greater effect on children than their household income.

Another interesting result is the difference according to the parent’s occupation. The significant results were derived especially in the white collar occupational group. In Korea, office work and drinking culture are inseparable. In Korean society, drinking alcohol is a type of nonverbal communication, which is believed to contribute greatly to maintaining bond formation that often occurs at the workplace [23,24] and it is not limited to men [17,51]. The reason it has a stronger outcome from maternal drinking frequency could be explained by the bond between a mother and a child.

The WHO states that well-being can be achieved when mental and physical health are balanced [52]. This approach to health is more critical for children in critical developmental stages that need full support from their parents [53,54]. In line with our findings, our study expresses the importance of the role of parents [54,55,56]. Depending on their parenting style, children can have differing outcomes in mental health when they are growing up [57,58]. Our study aimed to investigate the connection between parental drinking frequency and their children’s level of stress. Our study indicates that parents who drink frequently can cause harm to their children. These results suggest several tactics to lower stress levels in children, with consideration for their gender, age, household income, and parent’s occupation. Furthermore, we found that in order to lower the stress of children, modification on the working environment of parents is needed so that they can be focused on childcare, which can reduce a child’s stress.

### Limitations

Our study has several limitations that need to be considered in future studies. First, cross-sectional studies are inherently limited, so we could not identify causal relationships between the variables. Second, our study was conducted by grouping data into family units. Therefore, if a single member of the family did not provide a recordable answer, the entire family data set was censored, which led to considerable data reduction. To overcome the data reduction, we collected data from a period of 10 years, which provided adequate data for the analysis. Third, because we focused on both the parents’ influence on children, we censored families with one parent. This may have increased type 1 error in the study because our data group is limited to family groups in which both parents replied to the survey in its entirety. Fourth, we did not adjust for confounding variables. Finally, our main variable could present with more accurate measurement. We categorized based on the survey responses which did not have a unified unit of measure yet tried to provide a degree of alcohol consumption by categorizing into three groups.

## 5. Conclusions

Our study has focused on parents’ alcohol consumption and its impact on their children’s stress level. We discovered that parental drinking frequency negatively impacts stress in the children of drinkers. We suggest providing support care for children in vulnerable environments to reduce their stress levels. Policymakers should use our findings to build programs for adolescents’ mental health care, especially those who are living with heavy-drinking parents.

## Figures and Tables

**Table 1 ijerph-17-00257-t001:** General characteristics of study population.

Variables		Total (*N* = 3017)		Stress				*p*-Value
				Yes		No		
		N	(%)	N	(%)	N	(%)	-
**Maternal drinking frequency**		-		-	-	-	-	0.0004
	None	780	25.9	159	20.4	621	79.6	-
	Average	1899	62.9	491	25.9	1408	74.1	-
	High	338	11.2	104	30.8	234	69.2	-
**Paternal drinking frequency**		-		-	-	-	-	0.0006
	None	367	12.2	71	19.3	296	80.7	-
	Average	1319	43.7	309	23.4	1010	76.6	-
	High	1331	44.1	374	28.1	957	71.9	-
***Child***		-	-	-	-	-	-	-
**Sex**		-	-	-	-	-	-	-
	Boy	1589	52.7	395	24.9	1194	75.1	0.8584
	Girl	1428	47.3	359	25.1	1069	74.9	-
**Age group**		-	-	-	-	-	-	-
	Elementary school	505	16.7	119	23.6	386	76.4	0.6426
	Middle school	1392	46.1	357	25.6	1035	74.3	-
	High school	1120	37.1	278	24.8	842	75.2	-
**Depression**		-	-	-	-	-	-	-
	Yes	277	9.2	117	42.2	160	57.8	<0.0001
	No	2740	90.8	637	23.2	2103	76.8	-
**Self-assessment of health**		-		-	-	-	-	-
	Good	1969	65.3	432	21.9	1537	78.1	<0.0001
	Normal	895	29.7	260	29.1	635	70.9	-
	Bad	153	5.1	62	40.5	91	59.5	-
**Physical activity**		-	-	-	-	-	-	-
	No	86	2.9	23	26.7	63	73.3	0.7605
	Minimum	1495	49.6	376	25.2	1119	74.8	-
	Average	539	17.9	141	26.2	398	73.8	-
	Maximum	897	29.7	214	23.9	683	76.1	-
**Household Income**			-	-	-	-	-	-
	Q1 (Low)	173	5.7	60	34.6	113	65.3	0.0084
	Q2	661	21.9	175	26.5	486	73.5	-
	Q3	1054	34.9	243	23.1	811	76.9	-
	Q4 (High)	1129	27.4	276	24.4	853	75.6	-
***Mother***		-	-	-	-	-	-	-
**Age groups**		-	-	-	-	-	-	-
	30s	568	18.8	157	27.6	411	72.4	0.1069
	40s	2248	74.5	540	24.0	1708	76.0	-
	Above 50	201	6.7	57	28.4	144	71.6	-
**Education**		-	-	-	-	-	-	-
	Elementary school	123	4.1	41	33.3	82	66.7	0.0798
	Middle school	234	7.8	64	27.4	170	72.6	-
	High school	1708	56.6	406	23.8	1302	76.2	-
	University	952	31.6	243	25.5	709	74.5	-
**Smoking**		-	-	-	-	-	-	-
	Smoker	128	4.2	51	39.8	77	60.2	0.0004
	Ex-smoker	66	2.2	16	24.2	50	75.8	-
	Non-smoker	2823	93.6	687	24.3	2136	75.7	-
**Occupation**		-	-	-	-	-	-	-
	White collar	1267	42.0	346	27.3	921	72.7	0.0465
	Pink collar	313	10.4	71	22.7	242	77.3	-
	Blue collar	1249	41.4	300	24.0	949	76.0	-
	Others	188	6.2	37	19.7	151	80.3	-
**Self-assessment of health**		-	-	-	-	-	-	-
	Good	1186	39.3	231	19.5	955	80.5	-
	Normal	1421	47.1	365	25.7	1056	74.3	-
	Bad	410	13.6	158	38.5	252	61.5	-
***Father***		-	-	-	-	-	-	-
**Age groups**		-	-	-	-	-	-	-
	30s	137	4.5	41	29.9	96	70.1	0.0733
	40s	2291	75.9	550	24.0	1741	76.0	-
	Above 50s	589	19.5	163	27.7	426	72.3	-
**Education**		-	-	-	-	-	-	-
	Elementary school	140	4.6	46	32.9	94	67.1	0.0373
	Middle school	295	9.8	86	29.2	209	70.8	-
	High school	1219	40.4	294	24.1	925	75.9	-
	University	1363	45.2	328	24.1	1035	75.9	-
**Smoking**		-	-	-	-	-	-	-
	Smoker	1860	61.7	489	26.3	1371	73.7	0.1103
	Ex-smoker	670	22.2	155	23.1	515	76.9	-
	Non-smoker	487	16.1	110	22.6	377	77.4	-
**Occupation**			-	-	-	-	-	-
	White collar	1576	52.2	377	23.9	1199	76.1	0.5399
	Pink collar	918	30.4	241	26.3	677	73.7	-
	Blue collar	488	16.2	126	25.8	362	74.2	-
	Others	35	1.2	10	28.6	25	71.4	-
**Self-assessment of health**		-	-	-	-	-	-	-
	Good	1298	43.0	339	26.1	959	73.9	0.2607
	Normal	1386	45.9	327	23.6	1059	76.4	-
	Bad	333	11.0	88	26.4	245	73.6	-
**TOTAL**		3017	-	754	25.0	2263	75.0	-

The results of χ^2^ tests to analyze frequencies of each categorical variable by stress.

**Table 2 ijerph-17-00257-t002:** Factors associated with children’s stress level.

Variables		Stress			
		OR	95% CI		
**Maternal drinking frequency**					
	None	1.00		-	
	Average	1.39	1.11–1.74		
	High	1.58	1.14–2.19		
**Paternal drinking frequency**					
	None	1.00		-	
	Average	1.21	0.89–1.66		
	High	1.45	1.06–1.99		
***Child***					
**Sex**					
	Boy	1.00		-	
	Girl	0.99	0.83–1.17		
**Age group**					
	Elementary school	1.00		-	
	Middle school	1.12	0.87–1.44		
	High school	1.02	0.77–1.34		
**Depression**					
	Yes	2.36	1.81–3.08		
	No	1.00		-	
**Self-assessment of health**					
	Good	1.00		-	
	Normal	1.46	1.21–1.77		
	Bad	2.00	1.40–2.87		
**Physical activity**					
	No	1.27	0.75–2.13		
	Minimum	1.07	0.88–1.31		
	Average	1.15	0.89–1.49		
	Maximum	1.00		-	
**Household Income**					
	Q1 (Low)	1.00		-	
	Q2	0.74	0.56–0.98		
	Q3	0.81	0.61–1.07		
	Q4 (High)	0.77	0.58–1.02		
***Mother***					
**Age group**					
	30s	1.00		-	
	40s	0.88	0.68–1.12		
	Above 50s	0.97	0.61–1.52		
**Education**					
	Elementary school	0.87	0.52–1.45		
	Middle school	0.77	0.51–1.15		
	High school	0.85	0.68–1.06		
	University	1.00		-	
**Smoking**					
	Smoker	1.69	1.12–2.53		
	Ex-smoker	0.87	0.48–1.58		
	Non-smoker	1.00		-	
**Occupation**					
	White collar	1.69	1.12–2.54		
	Pink collar	1.10	0.68–1.78		
	Blue collar	1.34	0.88–2.02		
	Others	1.00		-	
**Self-assessment of health**					
	Good	1.00		-	
	Normal	1.38	1.13–1.68		
	Bad	2.45	1.89–3.17		
***Father***					
**Age group**					
	30s	1.00		-	
	40s	1.01	0.65–1.56		
	Above 50	1.27	0.77–2.09		
**Education**					
	Elementary school	1.00		-	
	Middle school	0.97	0.61–1.55		
	High school	0.80	0.52–1.23		
	University	0.83	0.52–1.32		
**Smoking**					
	Smoker	1.11	0.86–1.44		
	Ex-smoker	1.00	0.74–1.34		
	Non-smoker	1.00		-	
**Occupation**					
	Professional and skilled workers	0.76	0.34–1.71		
	Agriculture workers and technicians	0.82	0.37–1.84		
	Unskilled workers and non-workers	0.80	0.36–1.80		
	Others	1.00		-	
**Self-assessment of health**					
	Good	1.00		-	
	Normal	0.84	0.70–1.01		
	Bad	0.88	0.66–1.18		

**Table 3 ijerph-17-00257-t003:** Subgroup analysis of the association with maternal drinking frequency stress level.

		Maternal Drinking Frequency				
		None	Average		High	
		Adj. OR	Adj. OR	95% CI	Adj. OR	95% CI
**Child**						
**Sex**						
	Male	1.00	1.39	1.02–1.88	1.45	0.92–2.29
	Female	1.00	1.44	1.03–2.00	1.77	1.09–2.89
**Age groups**						
	Elementary	1.00	0.95	(0.53–1.68)	1.44	0.64–3.23
	Middle	1.00	1.38	(0.99–1.94)	1.46	0.88–2.39
	High	1.00	1.44	(1.00–2.07)	1.57	0.91–2.72
**Household Income**						
	Q1 (Low)	1.00	2.80	(1.61–4.87)	4.05	1.69–9.74
	Q2	1.00	1.13	(0.72–1.78)	1.03	0.52–2.02
	Q3	1.00	1.14	(0.75–1.75)	1.51	0.82–2.78
	Q4 (High)	1.00	1.51	(0.96–2.37)	2.02	1.04–3.92
**Occupation (Maternal)**						
	White collar	1.00	1.69	(1.17–2.44)	2.38	1.42–3.98
	Pink collar	1.00	1.28	(0.56–2.90)	0.44	0.13–1.45
	Blue collar	1.00	1.29	(0.92–1.81)	1.48	0.87–2.51
	Others	1.00	1.88	(0.58–6.12)	1.32	0.19–9.24

**Table 4 ijerph-17-00257-t004:** Subgroup analysis of the association with paternal drinking frequency and stress level.

		Paternal Drinking Frequency								
		None	Average				High			
		Adj. OR	Adj. OR	95% CI			Adj. OR	95% CI		
**Child**										
**Sex**										
	Male	1.00	1.27	(0.65–2.74)			1.16	0.42–3.25		
	Female	1.00	1.42	(1.12–1.80)			1.77	1.15–2.31		
**Age groups**										
	Elementary	1.00	0.97	(0.55–1.72)			1.44	0.64–3.23		
	Middle	1.00	1.37	(0.98–1.92)			1.46	0.88–2.39		
	High	1.00	1.50	(1.04–2.16)			1.57	0.91–2.72		
**Household Income**										
	Q1 (Low)	1.00	2.46	(1.51–4.00)			2.77	1.30–5.91		
	Q2	1.00	1.16	(0.73–1.85)			1.27	0.64–2.48		
	Q3	1.00	1.15	(0.74–1.80)			1.47	0.78–2.76		
	Q4 (High)	1.00	1.46	(0.92–2.32)			1.89	0.95–3.78		
**Occupation (Paternal)**										
	White collar	1.00	1.30	(0.94–1.80)			2.16	1.34–3.47		
	Pink collar	1.00	1.23	(0.82–1.84)			0.80	0.43–1.48		
	Blue collar	1.00	2.01	(1.11–3.65)			2.01	0.87–4.65		
	Others	1.00	–	–	–	–	–	–	–	–

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
