# Peer review of "Alcohol Consumption Frequency of Parents and Stress Status of Their Children: Korea National Health and Nutrition Examination Survey (2007–2016)"

_ijerph, 2019, doi:10.3390/ijerph17010257_

Round 1

Reviewer 1 Report

Kim and colleagues aimed to assess the associations between parental drinking patterns and stress of the children using Korean survey data. The study fits the scope of the journal and has the potential to attract the attention of readers. Unfortunately, the manuscript has several flaws, both in terms of study design and scientific communication, which prevent me from recommending it to publication in IJERPH. My concerns over the study are expressed below:

1. The use of scientific English is not appropriate and someone with a much better command of language should thoroughly edit the text.

Introduction

2. p. 1, line 38: It is not clear for me what is '...median regional mortality rate...'. This should be clarified.

3. p. 2, line 46: The statement that 'Koreans tend to drink more than people in other countries...' is very general and uninformative. According to recent OECD data, there are several member states of this organization where avg. consumption is higher. Obviously, several other states have lower rates of consumption. Still, you should provide some figures and references to make your argument convincing. By the way, the whole sentence shows a bit weird reasoning and I actually cannot see any reasonable logic in it.

4.p. 2, lines 62-63: You state that there are few studies on the topic and write nothing more about them; you neither give any details on these studies' results nor even provide references. This makes the research gap statement missing. Also, you do not explicitly show the original contribution of your study.

Methods

5. It is not clear to me how the selection of participants was conducted. You state that 'The initial study population was selected from children's dataset...'; but mention anything about how this 'selection' was done. What is totally unclear for me is how 3,796 children can be linked to 22,418 mothers and 16,347 fathers. How can the number of mothers be 6-fold higher than the number of their children? Clearly, I assume that I am wrong, but... you should explain what these number in fact mean and how to interpret them.

6. Why did not you use any stress scales but relied on your own stress indicator?

7. 'Interesting variable of main interest' is a bit awkward wording.

8. The way your main variable (line 84: frequency of drinking) is defined is faulty. Note that a person drinking 4, 5 or 6 times in a month cannot be assigned either to class '2' (less than 4 times in a month) or to class '3-6' (more than twice a week).

9. Moreover, your heaviest drinking class has a label of '3-6' which is further confusing. Are these three different classes and the ascending number refers to a higher frequency of drinking? Your results tables suggest the answer 'no', but what is the purpose of labelling it '3-6'? It seems to has no purpose and do nothing but confuse a reader.

10. What is the measure of physical activity used? This is not specified at all.

11. What are the occupation categories used? You state nothing about it in your methods section. Further in the results, you use two different sets of categories which is confusing (table 2: white, pink and blue-collar and others for mothers while for fathers you use: professional..., agriculture..., unskilled..., and others)! Why?

12. There is no description of model-building strategy at all.

13. You do not use any measures of goodness of fit for your models.

14. Instead of grouping five stress categories into a dichotomous variable, you could well use multinomial logistic regression making your results more robust in terms of the statistical method used.

15. There is no ethics statement.

Results:

16. Why there is such a massive disproportion of male and female children in your sample? This shows that something might have been wrong with your selection procedure. By the way, as I noted before, you do not provide too many details on the procedure.

17. You do not explain what is the purpose of your subgroup analysis.

18. p. 5, line 122: '...who did not drink frequently...' is incorrect as the ORs given above (1.21 and 1.39) are compared to reference category of non-drinkers and not non-frequent drinkers.

19. p. 5, line 125: there is no value of the upper bound of CI.

20. The heading 'High stress cognition' in your tables seems to be incorrect and should be 'Stress level' because 'high' is only one of the categories of your variable of interest.

Discussion:

21. Comparison of your findings with other studies is not appropriate. After discussing your results you simply state that they are in line with other studies but do not provide any details on this accordance. This is not enough and you should explicitly show how your findings comply with others'.

To conclude, sorry to write it but I cannot recommend the manuscript for publication in a reputable journal. In my opinion, the whole study needs extensive changes, including the study design itself.

Author Response

Thank you for reviewing our manuscript.

We appreciate your comments and for providing us with recent studies. We have added your suggested articles in our revised manuscript in the discussion section. We believe those will enhance our findings in our study.

Again, thank you for your kind support in our manuscript.

We wish to publish our manuscript with the International Journal of Environmental Research and Public Health.

Wonjeong Chae (on behalf of Dr. Jang)

Reviewer 1.

Kim and colleagues aimed to assess the associations between parental drinking patterns and stress of the children using Korean survey data. The study fits the scope of the journal and has the potential to attract the attention of readers. Unfortunately, the manuscript has several flaws, both in terms of study design and scientific communication, which prevent me from recommending it to publication in IJERPH. My concerns over the study are expressed below:

1.
The use of scientific English is not appropriate and someone with a much better command of language should thoroughly edit the text.

Response: Thank you for reviewing our manuscript. We received the editing service from the Bioscience Writers (the editing certificate is attached among submitted files). Also, we conducted the second editing through a native English speaker who has a master’s degree in Public Health from the University of Utah.

Introduction
2.
p. 1, line 38: It is not clear for me what is '...median regional mortality rate...'. This should be clarified.

Response: Thank you for your comment and we apologize for the unclear definition. We are afraid that you might misread the consumption rate to mortality rate. We provided the alcohol consumption rate of Korean adults. The data (figure) can be found in the Korea Center for Disease Control Regional Health Statistics. We rephrase it as “monthly alcohol consumption grouped by region” from “the median regional monthly alcohol consumption”. Also, we added the recent data of 2017 from Korean Statistics. We marked in red on page 1, line 40.

3.
p. 2, line 46: The statement that 'Koreans tend to drink more than people in other countries...' is very general and uninformative. According to recent OECD data, there are several member states of this organization where avg. consumption is higher. Obviously, several other states have lower rates of consumption. Still, you should provide some figures and references to make your argument convincing. By the way, the whole sentence shows a bit weird reasoning and I actually cannot see any reasonable logic in it.

Response: Thank you for your valuable comment. After reviewing your comment and our manuscript, we agree that our statement was not convincing that should be revised and provided with further information. Korea is not the highest alcohol consuming country in the world however, it is the highest country in Asia. WHO report in 2014 stated that Korea consumes 12.3 liters per year which is the highest in Asia and 2016 report (which was updated in March 2018) stated that Korea consumes 10.2 liters per year. It is higher than the global average of 6.3and regional average of 7.4. Also, we revised our statement on page 2, line 6-13.

4.
p. 2, lines 62-63: You state that there are few studies on the topic and write nothing more about them; you neither give any details on these studies' results nor even provide references. This makes the research gap statement missing. Also, you do not explicitly show the original contribution of your study.

Response: Thank you for your comment. We have revised that section to provide the purpose of this study and what we contribute from the study. Also, we added references as well. Please look into page 2, lines 33-38. Previously there were studies on the relationship between the mental health of children and alcoholic parents. However, since 1988, no study has the focus was on the stress level of children/adolescents with parents who are heavy drinkers. Also, as stress and mental health are closely related, we believe controlling a risk factor such as stress level can be beneficial to maintaining the health status of children. 

Methods:
5.
It is not clear to me how the selection of participants was conducted. You state that 'The initial study population was selected from children's dataset...'; but mention anything about how this 'selection' was done. What is totally unclear for me is how 3,796 children can be linked to 22,418 mothers and 16,347 fathers. How can the number of mothers be 6-fold higher than the number of their children? Clearly, I assume that I am wrong, but... you should explain what these number in fact mean and how to interpret them.

Response: Thank you for reviewing our manuscript. We apologize that there was a lack of explanation about the structure of the data. Among the total, data under 19 years old were named “Child” It seems that naming the 'women over 30' and 'men over 30' group as 'maternal' and 'paternal' could confuse. The child has a unique family ID, which is the same as the parent of that child. Using this, we chose “adult women” and “adult men” to connect with the child, and removed the rest. After the selection, we defined adult women who matched 3,796 children as “maternal” and adult men as “paternal”. We revised our manuscript to be clearer to understand. Please see section 2 Methods, 2.1 study design, and participants, page 2 line 48 – page 3 line 2.

6.
Why did not you use any stress scales but relied on your own stress indicator?

Response: Thank you for your comment. For the study, we did not use scales instead we measured stress level by the response of the survey question from K-NHANES's standard of stress. The survey asked subjects how much stress they felt. Based on the answer, we grouped them into “Yes” (very often, often) and “No” (occasionally and a few times). We inserted the explanation for clarity. Please see the section methods, 2.2 dependent variables, page 3, lines 6-9. Also, we have explained more on your comment regarding the reason we conducted binomial logistic regression instead of multinomial logistic. In the manual for the K-HANES, there is the stress status variable based on the same survey question we chose for the study and it guided us to group them into two categories. 

7.
'Interesting variable of main interest' is a bit awkward wording.

Response: Thank you for your comment. We revised it to “Interesting Variable”.

8.
The way your main variable (line 84: frequency of drinking) is defined is faulty. Note that a person drinking 4, 5 or 6 times in a month cannot be assigned either to class '2' (less than 4 times in a month) or to class '3-6' (more than twice a week).

Response: Thank you for your valuable comment. We apologize for not providing details in our grouping structure and presenting poorly on the manuscript. It was also defined according to the K-NHANES survey question responses. K-NHANES responded to the frequency of drinking into six scales: 1) never, 2) less than once a month, 3) once a month, 4) 2-4 times a month, 5) 2-3 times a week, and 6) more than 4 times a week. Therefore, we divided it into the following three groups to highlight the difference: None (never), Average (less than 4 times in a month and once a month) and High with remaining categories. However, after reviewing your comment, we agree that it could cause some confusion on the classification of our main variable that we added in the methods, 2.3 interesting variable, page 3, line 12-17 and the limitation section page 11, line 4-7. 

9.
Moreover, your heaviest drinking class has a label of '3-6' which is further confusing. Are these three different classes and the ascending number refers to a higher frequency of drinking? Your results tables suggest the answer 'no', but what is the purpose of labelling it '3-6'? It seems to has no purpose and do nothing but confuse a reader.

Response: Thank you for reviewing our manuscript. Again, we apologize for not providing details and presenting poorly on our manuscript. We have revised the section to be easier for readers to understand our variable and its categories. Please see the methods, 2.3 interesting variable, page 3, lines 12-17.

10.
What is the measure of physical activity used? This is not specified at all.

Response: Thank you for reviewing our manuscript. Physical activities include walking and weight exercise. Again, those categories are based on the survey question that we grouped them into four. No means no physical activities; Minimum means walking or weight exercise 1-2 times a week; Average means walking or weight exercise 3-4 times a week; Maximum means walking or weight exercise 5-7 times a week. We have added on page 3 lines 23-25 in 2.4 covariates section.

11.
What are the occupation categories used? You state nothing about it in your methods section. Further in the results, you use two different sets of categories which is confusing (table 2: white, pink and blue-collar and others for mothers while for fathers you use: professional..., agriculture..., unskilled..., and others)! Why?

Response: Thank you for your valuable comment. We apologize for the inconsistent use of terminology that caused you a huge confusion. There was an error in that we wrote the manuscript. When we performed the analysis again to fulfill your below comment, we checked the variable for occupation and unified for both parents. We modified the names of the mother’s and father’s variables marked differently. It was classified by the same criteria, but the error appears to have occurred during the writing of the paper. The occupational groups were divided into three groupswhite-collar (administrative and management setting), blue-collar (manual labor industry) and pick-collar (service industry), and others (student, unemployed, homemakers). Please find page 3, lines 33-35, in 2.4 covariates section.

12.
There is no description of model-building strategy at all.

Response: Thank you for your valuable comment. For the study, we chose the top-down method that we set up the hypothesis on the stress level of children and their parent’s drinking frequency. After that, we included covariate variables to conduct the analysis.

13.
You do not use any measures of goodness of fit for your models.

Response: Thank you for your comment on our manuscript. After your comment, we have conducted the Hosmer and Lemeshow Goodness Test to evaluate the goodness of the binary logistic model in Table 2 and 3 analysis, and we have added and attached values. In the model test, there were some non-significant models in table 3 and this was excluded from the analysis. Please see the page 3, lines 40-43 in 2.5 statistical analyses. As the results, our model showed chi-square value of 5.7471, p-value: 0.6755.

14.
Instead of grouping five stress categories into a dichotomous variable, you could well use multinomial logistic regression making your results more robust in terms of the statistical method used.

Response: Thank you for your valuable comment. We apologize that there was an error in our manuscript that we described wrongly. The survey question that we used was “How much stress do you usually feel?” (survey question variable number: BP1) and it has four responses: 1) I feel so much 2) I feel a lot, 3) I feel a little bit, and 4) I hardly feel stress. In the manual of K-NHANES, they show to calculate the stress cognitive rate based on the same variable we have used for the study. It shows that 1 and 2 consider as stressed and 3 and 4 consider as not stressed. Therefore, we followed the K-NHANES’s guideline to measure stress and analyzed with binominal logistic regression. Please see the page 3, lines 12-17, 2.3 interesting variable for revised description. 

15.
There is no ethics statement.

Response: Thank you for reviewing our manuscript. The study is conducted by secondary data and it has been deidentified for the survey participants. This survey data is from the Korea Centers for Disease Control and Prevention as open data that any researcher can utilize for their study to enhance health of Korean population. The survey was approved by the Institutional Review Board (IRB) of the Korea Centers for Disease Control and Prevention For the revised manuscript, we inserted the ethical statement on page 4, lines 1-5.

Results:
16.
Why there is such a massive disproportion of male and female children in your sample? This shows that something might have been wrong with your selection procedure. By the way, as I noted before, you do not provide too many details on the procedure.

Response: Thank you for valuable comments. We reviewed our analysis and found an error in the process. We conducted an analysis from the beginning and entered new tables. Due to the changes in table 1, we also changed table 2 and 3 as well. Also based on your comment, we tried to provide our procedure in the method section. Please see page 3, lines 40-43.

17.
You do not explain what is the purpose of your subgroup analysis.

Response: Thank you for reviewing our manuscript. We are sorry that we have not discussed the purpose of the subgroup analysis. As we chose the top-down approach, we wanted to investigate factors that could cause higher stress levels among children who have high alcohol-consuming parents. As a result, we found that between mother and father, mothers influence more on the child. As per socioeconomic factors, lower incomed family child and white-collar parent’s child had higher stress when their parents consume higher alcohol. We added this with further description on page 9, lines 27-30 in the discussion.

18.
p. 5, line 122: '...who did not drink frequently...' is incorrect as the ORs given above (1.21 and 1.39) are compared to reference category of non-drinkers and not non-frequent drinkers.

Response: Thank you for reviewing our manuscript. We revised the line correctly while reviewing writings for results.

19.
p. 5, line 125: there is no value of the upper bound of CI.

Response: Thank you for your comment on our manuscript. We have revised the line and provided the upper CI figure in the revised manuscript. 

20.
The heading 'High stress cognition' in your tables seems to be incorrect and should be 'Stress level' because 'high' is only one of the categories of your variable of interest.

Response: Thank you for your comment. We agree that it should be revised to the stress level and revised accordingly.

Discussion:

21.
Comparison of your findings with other studies is not appropriate. After discussing your results you simply state that they are in line with other studies but do not provide any details on this accordance. This is not enough and you should explicitly show how your findings comply with others'.

Response: Thank you for your valuable comment. We revised the discussion and provided further information regarding previous studies and compared those findings with our study. Please see page 10, lines 8-32 in the discussion.

Reviewer 2 Report

This is a very interesting paper, drawing on a large scale survey to draw conclusions about links between parental drinking and chidlren's stress. I only have a few suggstions for amendments. The introducion has a few English issues and should be reviewed - lines 32, 33, 34.

Content-wise, a few questions/suggestions: would be good to have consietnt and clear definition of terms such as 'drinking problems' and 'normal'.  Line 59 introduces the concept of 'alcoholic', which is different from the rest of the discussion.

Lines 46 and 47: I would like some explanation of this cultural link, with evidence to support this.

I thought that in places the discussion section could possibly be more analytical. Aside from - line 179 - suggesting more investigation is required, what would you hypothesise re difference in different populations re the relationships between, fathers, mothers and children?

Good that you are aware of the limitations of the paper, which I agree with. However, this paper makes an important contribution to understanding the patterning of harms to children in Korea due to parental drinking.

Author Response

Thank you for reviewing our manuscript.

We appreciate your comments and for providing us with recent studies. We have added your suggested articles in our revised manuscript in the discussion section. We believe those will enhance our findings in our study.

Again, thank you for your kind support in our manuscript.

We wish to publish our manuscript with the International Journal of Environmental Research and Public Health.

Wonjeong Chae (on behalf of Dr. Jang)

Review Reviewer 2

Comments and Suggestions for Authors

1.
This is a very interesting paper, drawing on a large scale survey to draw conclusions about links between parental drinking and chidlren's stress. I only have a few suggstions for amendments. The introduction has a few English issues and should be reviewed - lines 32, 33, 34.

Response: Thank you for your interest in our manuscript and after receiving your comment, we did further editing for the manuscript with a native English speaker who has a master’s in Public Health at the University of Utah. Also, we attached the editing certificate for BioScience Writers, a professional editing company. 

Content wise, a few questions/suggestions: 

2.
Would be good to have consistent and clear definition of terms such as 'drinking problems' and 'normal'.  Line 59 introduces the concept of 'alcoholic', which is different from the rest of the discussion.

Response: Thank you for your comment. We do agree that the terminology should be consistent that we revised to our manuscript with further explanation. We used the term alcoholic because that was used in the previous study. However, after reviewing the manuscript again and we decided to use “parents who consume a high volume of alcohol” instead of alcoholics. 

3.
Lines 46 and 47: I would like some explanation of this cultural link, with evidence to support this.

Response: We appreciate your valuable comment. We did not provide a full explanation of the linkage between occupation and alcohol consumption in the aspect of Korean culture. We explained in the manuscript in the introduction, page 2, lines 14-20.

4.
I thought that in places the discussion section could possibly be more analytical. Aside from - line 179 - suggesting more investigation is required, what would you hypothesise re difference in different populations re the relationships between, fathers, mothers and children?

Response: Thank you for your comment that can improve our manuscript. We agree that further investigation is needed that we conducted subgroup analysis by household income and occupation (maternal). Also, we described the findings in the discussion section page 10, lines 16-32. From the analyses, we found that children with lower household income, mothers who have white collar jobs have higher stress. We explained the outcome of analysis related to the cultural aspect as well.

5.
Good that you are aware of the limitations of the paper, which I agree with. However, this paper makes an important contribution to understanding the patterning of harms to children in Korea due to parental drinking.

Response: We deeply appreciate all of your comments that can improve our manuscript. We hope our study can use to providing mental health care for children.

Reviewer 3 Report

Summary

The stated goal is to evaluate the relationship between parental drinking and childhood stress.  The results are interesting but i do have some questions about statements in the introduction, the methodology used to assess drinking, depression, etc, as well as additional analyses that could increase the impact of the manuscript.

Overall

This manuscript will benefit from editing by a native English speaker

Introduction

Can additional information be given on alcohol consumption rate (lines 37-41)?  Likewise, information about alcohol consumption by men v women would be of interest.  Since it is mentioned that people in Korea drink more than other countries (lines 45-46) I am especially curious to see how this compares to rates in other parts of the world. Lines 51-52. "Stress is not only a problem for adults; adolescents also suffer from the negative impacts of stress and anxiety and their growth [24-26]." Growth of what?

Methods

Lines 74-79.  Was a validated test for childhood stress used?  If so what is the name.  Was only a single question used?  If so, has this been validated or published as an established method to assess childhood stress? Lines 80-84.  Was a validated alcohol consumption frequency used?  If so what is the name.  Was only a single question used?  If so, has this question been validated or published as an established method to assess alcohol consumption across the previous year? What is “high stress cognition”?  This does not seem to be defined in the methods.  Should it be high stress condition? Other methodological questions.  How is depression defined/assessed?  How is physical activity assessed?  What are the income brackets?  Why are the occupation categories for mother and father different?  What is a pink color worker?  Should an unskilled worker really be grouped together with a non-worker?

Results

There are some formatting issues with the tables (e.g., Child Sex?). Line 125.  CI needs to be added.  Right now it only says 0.88. It was mentioned that statistically significant odds ratios but no P values have been included.  P values should be included to support the statement. Perhaps some type of model could be developed that would predict which factors are mostly likely to result in high stress in children (e.g., maternal drinking in high school aged female children).

Discussion

Line 160.  “(refs)” needs to be corrected.

Author Response

Thank you for reviewing our manuscript.

We appreciate your comments and for providing us with recent studies. We have added your suggested articles in our revised manuscript in the discussion section. We believe those will enhance our findings in our study.

Again, thank you for your kind support in our manuscript.

We wish to publish our manuscript with the International Journal of Environmental Research and Public Health.

Wonjeong Chae (on behalf of Dr. Jang)

Reviewer 3

Overall

This manuscript will benefit from editing by a native English speaker

Response: Thank you for reviewing our manuscript. Upon your comment, we received a native English speaker’s editing who has a master’s degree in Public Health. Also, we attached editing certificate from the BioScience Writer Company after the service.

Introduction

1.
Can additional information be given on alcohol consumption rate (lines 37-41)?  Likewise, information about alcohol consumption by men v women would be of interest.  Since it is mentioned that people in Korea drink more than other countries (lines 45-46) 

I am especially curious to see how this compares to rates in other parts of the world. Lines 51 -52. 

"Stress is not only a problem for adults; adolescents also suffer from the negative impacts of stress and anxiety and their growth [24-26]." Growth of what?

Response: Thank you for reviewing our manuscript and your valuable comments. We revised the manuscript and provided WHO health data in the introduction, page 2, line 14-20 that Korea is the highest country in Asia who consume a high volume of alcohol. Also, on line 52, it was the development related to the cognitive level. Please see page 2, line 25 for clarification.

Methods

2.
Lines 74-79.  
Was a validated test for childhood stress used?  If so what is the name.  
Was only a single question used?  If so, has this been validated or published as 
an established method to assess childhood stress?

Response: Thank you for your comment. For the study, we did not use scales instead we measured stress level by the response of the survey question from K-NHANES's standard of stress. The survey asked subjects how much stress they felt. Based on the answer, we grouped them into “Yes” (very often and often) and “No” (occasionally and a few times). We inserted the explanation for clarity. Please see the section methods, 2.2 dependent variable, page 3, lines 6-9. Also, we have explained more on your comment regarding the reason we conducted binomial logistic regression instead of multinomial logistic. In the manual for the K-HANES, there is the stress status variable based on the same survey question we chose for the study and it guided us to group them into two categories. 

3.
Lines 80 -84.  
Was a validated alcohol consumption frequency used?  If so what is the name.  
Was only a single question used?  If so, has this question been validated or published as an established method to assess alcohol consumption across the previous year? 

Thank you for your valuable comment. We apologize for not providing details in our grouping structure and presenting poorly on the manuscript. It was also defined according to the K-NHANES survey question responses. K-NHANES responded to the frequency of drinking into six scales: 1) never, 2) less than once a month, 3) once a month, 4) 2-4 times a month, 5) 2-3 times a week, and 6) more than 4 times a week. Therefore, we divided it into the following three groups to highlight the difference: None (never), Average (less than 4 times in a month and once a month) and High with remaining categories. However, after reviewing your comment, we agree that it could cause some confusion on the classification of our main variable that we added in the methods, 2.3 interesting variable, page 3, line 12-17 and the limitation section page 11, line 4-7. 

4.
What is “high stress cognition”?  This does not seem to be defined in the methods.  Should it be high stress condition? Other methodological questions.  How is depression defined/assessed?  

Response: Thank you for reviewing our manuscript. Regarding the definition of stress, we followed the K-NHANES guideline. And explained in detail in the above comment. After reviewing your comment and manuscript, we changed the high-stress cognition to the stress level. It was our mistake with translating terms from Korean to English that caused confusion. Depression was defined based on the survey question: “have you feel depressed in the past 2 weeks?” 

5.
How is physical activity assessed?  

Response: Thank you for reviewing our manuscript. Physical activities include walking and weight exercise. Again, those categories are based on the survey question that we grouped them into four. No means no physical activities; Minimum means walking or weight exercise 1-2 times a week; Average means walking or weight exercise 3-4 times a week; Maximum means walking or weight exercise 5-7 times a week. We have added on page 3 lines 23-25 in 2.4 covariates section.

6.
What are the income brackets?  

Response: Thank you for your comment. For the household income, we used the survey question and its responses are in four categories. Participants can choose from high, mid-high, mid-low, and low. We regret to inform you that we are not able to provide the range of each category. 

7.
Why are the occupation categories for mother and father different?  What is a pink color worker? Should an unskilled worker really be grouped together with a non-worker?

Response: Thank you for your valuable comment. We apologize for the inconsistent use of terminology that caused you a huge confusion. There was an error in that we wrote the manuscript. We modified the names of the mother’s and father’s variables that marked differently. It was classified by the same criteria, but the error appears to have occurred during the writing of the paper. One reviewer recommended us to check the goodness of fit and while we conducted the analysis, we checked the occupational variable again. The occupational groups were divided into three groupswhite-collar (administrative and management setting), blue-collar (manual labor industry) and pick collar (service industry), and others (student, unemployed, homemakers). Please find page 3, lines 33-35, in 2.4 covariates section.

Results

8.
There are some formatting issues with the tables (e.g., Child Sex?). Line 125.  
CI needs to be added.  Right now it only says 0.88. It was mentioned that statistically significant odds ratios but no P values have been included.  P values should be included to support the statement. Perhaps some type of model could be developed that would predict which factors are mostly likely to result in high stress in children (e.g., maternal drinking in high school aged female children).

Response: Thank you for reviewing our manuscript. We have reviewed and revised section results to provide all necessary information. Instead of the p-value, we provided 95% CI for all variables that we mentioned in our manuscript. 

Discussion
9.
Line 160.  “(refs)” need                                        

Response: Thank you for your valuable comment. We apologize that we didn’t provide appropriate references. We have added references on line 160 as you mentioned.